# A qualitative photo-elicitation study exploring the impact of falls and fall risk on individuals with subacute spinal cord injury

**Olinda D. Habib Perez[1], Samantha Martin[1,2], Katherine Chan[1], Hardeep Singh[1,3], Karen K. Yoshida[2,4,5], Kristin E. Musselman**[1,2,4]*

**1** KITE, Toronto Rehab–University Health Network, Toronto, ON, Canada, **2** Department of Physical Therapy, University of Toronto, Toronto, ON, Canada, **3** Bridgepoint Collaboratory for Research and Innovation, Lunenfeld-Tanenbaum Research Institute, Sinai Health System, Toronto, ON, Canada, **4** Rehabilitation Sciences Institute, University of Toronto, Toronto, ON, Canada, **5** Dalla Lana School of Public Health (Social Science Division), University of Toronto, Toronto, ON, Canada

* kristin.musselman@utoronto.ca

## Abstract

### Background

Individuals living with chronic spinal cord injury or disease (SCI/D) are at an increased risk of falling. However, little is known about the impact of falls and fall risk in the subacute phase of SCI/D, despite this being a time when fall prevention initiatives are delivered. Hence, we explored the impact of falls and fall risk in individuals with subacute SCI/D as they transitioned from inpatient rehabilitation to community living.

### Methods

This qualitative photo-elicitation study used an inductive thematic analysis. Eight individuals (7 male) undergoing inpatient rehabilitation at a Canadian tertiary rehabilitation hospital due to a new SCI/D participated. Six months following discharge, photo-elicitation interviewing was used to understand the impact of falls and fall risk. Over 7–14 days, participants completed a photo-assignment that involved taking photographs in response to questions, such as what increases/decreases your likelihood of falling? A semi-structured interview followed, in which participants described their photographs and discussed their experiences with falls, fall risk and fall prevention training.

### Results

Four themes were identified. 1) Risk factors and strategies identified through lived experience. Participants discovered their fall risk factors and fall prevention strategies through "trial and error". 2) Influences on the individual's perception of their fall risk. Prior experience with falls, including falls experienced by themselves as well as friends and family, influenced their perception of fall risk. 3) Experiencing life differently due to increased fall risk. A high fall risk reduced participation, increased negative emotions and decreased independence and quality of life. 4) Falls training in rehabilitation can be improved. Prior experiences with

**Data Availability Statement:** Data cannot be shared publicly because the data contain potentially identifying information. Data are available from the University Health Network Ethics Committee

(contact via email: reb@uhnresearch.ca) for researchers who meet the criteria for access to confidential data.

**Funding:** This research was funded by a grant from the Canadian Institute of Health Research (PTJ 153017) to KEM. https://cihr-irsc.gc.ca/e/193.html The funders had no role in study design, data collection and analysis, decision to publish, or preparation of the manuscript.

**Competing interests:** The authors have declared that no competing interests exist.

falls training varied; however, participants expressed a desire for comprehensive and individualized training.

## Conclusion

Although participants' experiences with falls and fall prevention varied, falls and the risk of falling can have a significant impact on the first year of living with a SCI/D.

## Introduction

Following spinal cord injury or disease (SCI/D), most individuals experience falls and a heightened risk of falling [1]. People living with SCI/D have described their fall risk as individualized and multifactorial, whereby each individual has a unique set of risk factors that may interact to cause a fall [2]. The circumstances surrounding falls among the SCI/D population have been well described in recent years [3], with circumstances varying with mobility status [4, 5]. Compared to wheelchair users with SCI/D, those who walk have a higher incidence proportion of falls [1] and experience the majority of their falls when walking or completing activities in standing [4, 6, 7]. Similarly, the physical, psychological and social consequences of falling and living with a high fall risk after SCI/D have been described [3, 6, 8–11]. In addition to the more visible consequences of a fall, such as injury, hospital admission and increased attendant care, individuals with SCI/D have also described significant psychosocial impacts, such as feelings of frustration, vulnerability and embarrassment, reduced work productivity and a restriction of meaningful recreational activities [9–11].

Our increased understanding of the causes and consequences of falls after SCI/D may translate to improvements in the delivery of fall prevention education and training for this population [12, 13]. However, our current knowledge of falls and the risk of falling after SCI/D is based on research involving individuals in the chronic phase of injury (i.e. one or more years post-injury onset) [1]. Little is known about the occurrence, causes and consequences of falls early after SCI/D. It is estimated that 7–20% of rehabilitation inpatients with SCI/D fall [14–16], while 69–78% of individuals living with chronic SCI/D fall each year [1]; hence, a higher fall incidence likely manifests after individuals with SCI/D have been discharged from inpatient hospital care. Indeed, individuals with SCI/D have reported mobility tasks, such as transferring and maintaining balance, to be one of the greatest challenges to community integration following hospital discharge [17]. Thus, the subacute period of SCI/D (i.e. within the first year post-injury), when individuals transition from inpatient rehabilitation to community living, is a critical time for fall prevention initiatives.

Currently, fall prevention education is delivered early after SCI/D through inpatient and outpatient rehabilitation; however, hospital administrators have noted a lack of consistency in the content and quantity of fall prevention education delivered [18]. Moreover, physical and occupational therapists in hospital environments have noted several challenges with delivering fall prevention education and training for their patients with SCI/D, including limited time due to short lengths of hospital stay and a lack knowledge about the fall risks that their patients experience in the community [19]. A greater understanding of falls and fall risk in the subacute period may translate to increased relevance of the fall prevention education and training provided during the limited time spent in inpatient and outpatient rehabilitation. As a first step toward more effective fall prevention education and training after SCI/D, we explored the impact of falls and fall risk during the transition from hospital to community living among individuals with SCI/D through photo-elicitation and one-on-one interviews.

## Methods

### Design

This was a qualitative, photo-elicitation [20–22] study that used an inductive thematic analysis [23]. Photo-elicitation was used because photographs not only provide visual data that can help researchers gain greater depth of understanding on an issue, but they also facilitate story telling among the participants, enabling a deeper reflection on their experiences [24]. Use of an inductive thematic analysis was suitable for the current study as it aimed to code data collected in qualitative interviews without the limits of a pre-existing coding framework [25].

This study was conducted at the Lyndhurst Centre, Toronto Rehab Institute-University Health Network (UHN) and received approval from the UHN Research Ethics Board. The current study reports the qualitative data from a larger research project exploring the causes and consequences of falls across the continuum of care (i.e., from inpatient rehabilitation to the community) in Canadians with subacute SCI/D. The consolidated criteria for reporting qualitative research (COREQ) was followed (see S1 Checklist).

### Setting and participants

Individuals undergoing inpatient rehabilitation at the Lyndhurst Centre were invited to participate through a centralized recruitment process [26]. The Lyndhurst Centre is a tertiary rehabilitation hospital dedicated to SCI/D that receives approximately 300 inpatient admissions and 20,000 outpatient visits per year [27]. Participants were included in the study if they: 1) were 18 years of age or older; 2) experienced a traumatic or non-traumatic and non-progressive SCI/D; 3) had an American Spinal Injury Association Impairment Scale rating of A-D; 4) were an inpatient in a rehabilitation hospital at the time of enrollment; and 5) did not have any other significant co-morbidities that affected balance or mobility. From the group of 60 participants that were recruited for the larger study, participants who lived within 100km of the Lyndhurst Centre were invited to participate in photo-elicitation approximately six months after discharge from inpatient rehabilitation. These individuals were sent a recruitment flyer through email or the postal service. Our desired sample size of 7–10 individuals was based on previous photo-elicitation research [11, 28, 29] and the sample size recommended for photo-based group discussion [30]. Participants of the current study engaged in a focus group meeting as part of the larger research project, and a sample size of 7–10 participants enabled in-depth group discussion [30].

### Data collection

Individual photo-elicitation interviews were used to collect information about the impact of falls and fall risk on the participants' lives during the transition to community living. When participants completed the photo-elicitation interview, they had been discharged from inpatient rehabilitation for at least six months. Prior to discharge, participants were introduced to two of the researchers (OHP and KC), who then contacted the participants monthly for six months following discharge. Participants were asked to complete a photo-assignment, which involved taking at least two photographs over seven days in response to three questions: (1) What increases your likelihood of falling?; (2) What decreases your likelihood of falling?; and (3) How does the risk of falling affect your ability to participate in recreational activities and/or work (paid or volunteer)? Participants were supplied with the following definition of a fall: an event that results in a person coming to rest inadvertently on the ground or floor of other lower level [31]. A researcher explained the photo-assignment to participants over the phone. This discussion also included a review of the guidelines for privacy and ethics when taking photos. Photo-elicitation data collection occurred between April 2019 and January 2020.

Semi-structured interviews were conducted within two weeks of completing the photo-assignment. Five interviews were conducted in person and three interviews were conducted over the phone, all lead by OHP (post-doctoral fellow, RKin, PhD in Rehabilitation Sciences). Interviews were approximately one hour, and participants were asked to describe 2–3 of their photos for each question on the photo-assignment and the reasons they chose to take them. The interviews included open-ended questions following the SHOWeD framework for photo-voice research. This framework involves a set of five open-ended questions: (1) What do you see here?; (2) What is really happening here?; (3) How does this relate to our lives?; (4) Why does this situation, concern, or strength exist?; and (5) What can we do about it? [32]. Additional interview questions concerning their experiences and opinions about fall prevention training in rehabilitation were asked. See S1 File for the interview guide. All interviews were recorded using a hand-held digital recorder.

### Data analysis

The audio-recorded interviews were anonymized and transcribed verbatim by a researcher (KC or OHP). The transcription of all interviews and thematic analysis was conducted based on Braun and Clark's approach to thematic analysis [25]. Four researchers (SM, OHP, KEM, and KC–all women) read all eight transcripts multiple times and individually made notes on ideas conveyed in the transcripts. The same researchers then discussed as a group and interpreted the data to generate initial ideas for codes. Codes represent basic elements or pieces of information about the studied phenomenon that are identified in the transcribed dialogue [33]. Through discussion, the four researchers reached agreement on an initial codebook (i.e. code labels with definitions). SM then applied this coding scheme to each transcript manually, discussing any edits or additions to the codebook as they arose with the other three researchers. SM and KEM created preliminary themes based on patterns identified in the list of codes. Codes that represented similar concepts were grouped together to form subcategories or categories, depending on the number of levels of hierarchy that emerged in the data. Themes were identified for categories that grouped together in a meaningful way. Final themes, categories, and subcategories were developed through an iterative process involving contribution from four researchers (SM, KEM, OHP, and KC). All six co-authors provided feedback and contributed to the development of the final set of themes, categories and subcategories at various stages of the analysis process. SM was a physiotherapy student completing a research internship through the University of Toronto and is currently a licensed physiotherapy resident. KEM is a Scientist with a background in physical therapy and qualitative research. KC is a Research Coordinator with a graduate degree in Rehabilitation Sciences and undergraduate degree in Kinesiology. The use of photos and direct quotes from participants helped to ensure that the interpretations were grounded in data. An audit trail of the analytic process was maintained to enhance transparency and credibility of the research. Two coauthors (KY and HS) who were not directly involved in the analysis reviewed the findings to enhance confirmability of the interpretations and decrease potential bias [34–36].

### Results

Forty-five individuals were invited to participate. Eight individuals (seven male, one female) with non-traumatic SCI/D in the subacute stage (i.e. mean ± standard deviation time post-injury = 9.0 ± 1.0 months) participated. Ambulatory status varied across the participants, with seven participants ambulating as their primary means of mobility and one using a manual wheelchair at the time of the interview. Participant demographics are presented in Table 1. Three participants changed their ambulatory status between discharge from inpatient

**Table 1. Participant characteristics.**

| Participant code | Sex | Mobility status | 5-year age category | Neurological level of injury | AIS | Number of falls |
|---|---|---|---|---|---|---|
| F09 | M | Ambulated with cane/walking poles | 55–59 | Lumbar | C | 3 |
| F11 | M | Ambulated with walker | 70–74 | Lumbar | D | 0 |
| F20 | M | Ambulated with cane | 70–74 | Cervical | D | 0 |
| F30 | M | Ambulated without gait aid | 50–54 | Cervical | D | 1 |
| F40 | M | Ambulated with walker/cane | 30–34 | Cervical | D | 1* |
| F44 | M | Ambulated with walker | 70–74 | Cervical | D | 4 |
| F56 | F | Used manual wheelchair | 55–59 | Lumbar | A | 0 |
| F74 | M | Ambulated without gait aid | 65–69 | Cervical | D | 0 |

F, female; M, male; AIS, American Spinal Cord Injury Association Impairment Scale rating at admission to inpatient rehabilitation. Mobility status reported is the status at the time of the interview. Number of falls is the number of falls experienced during the first six months post-discharge from inpatient rehabilitation.

*F40 experienced an additional fall while on a weekend pass prior to inpatient discharge.

rehabilitation and the end of the study: F11 and F44 switched from using a manual wheelchair to a walker; F30 switched from using a cane to ambulating without a gait aid.

Four main themes were revealed following qualitative analysis of the data: (1) Risk factors and strategies identified through lived experience; (2) Influences on the individual's perception of their fall risk; (3) Experiencing life differently due to increased fall risk; and (4) Falls training in rehabilitation can be improved (see Table 2 and Fig 1).

## Theme 1: Risk factors and strategies identified through lived experience

Through their own experiences, participants realized their fall risk factors and developed various strategies to mitigate risks. These strategies were specific to the individual, as well as to the situation, and were described as being developed through "trial and error." Some strategies were developed after experiencing near falls: "When I walk down [a step] and I almost tripped, and I say, 'Wow, I better pay more attention'" (F11). Participants also developed strategies based on previous falls, which allowed them to realize how to recognize potential risks in the environment. For example, F20 stated: "From experience. . .I had fallen before because of irregularities in the levels of the sidewalk."

**Category 1a: Biological & behavioural factors can increase risk of falling.** Participants discussed biological and behavioral factors that increased their risk of falling. Biological factors were physical impairments resulting from the SCI/D were also discussed and included balance impairments, impaired gait, limited strength, fatigue, and muscle spasms. A couple of participants considered how their "condition varies from day to day" (F30) or depends on what they did the day before. For example, F09 explained, "Today I'm a little stiffer than normal mostly because I had a good workout yesterday. So, when I try to do things that I could have done yesterday safely. . .I could hurt myself." Behavioral factors included inattention, overconfidence, and nervousness. F20 described how being excited took away from their ability to pay attention to the fall risks in the environment: "I mean shopping is a great experience, so you're excited and you're enjoying yourself, and often I shop with other people, and so it's a social occasion and so I'm not paying as much attention." F20 described how being nervous increased their risk of falling: "When there are a lot of stairs and I come to the top of them, and I look down, first of all it makes me nervous, which increases my likelihood of falling."

**Category 1b: The environment is often less accessible to individuals with SCI/D and increases fall risk.** Many of the risk factors described by participants were part of the environment, including weather, city infrastructure, public spaces, and their homes. These

**Table 2. Themes, categories, subcategories, and supporting quotes.**

| |
|---|
| **Theme 1: Risk factors and strategies identified through lived experience** |
| Category 1a: Biological and behavioral factors can increase risk of falling |
| (i) "Yeah, my balance is completely messed up. And, just a lot of what I do, with my injury, I have a fear of falling when I do yoga".—F30, ambulator |
| (ii) "I see the bottom of the step, which for some reason always increases my fatigue. . . the last time I fell, I was actually good coming down the steps. And, then I got to the bottom and as soon as I stepped onto the floor, my leg just stopped, gave out."—F40ambulatory |
| Category 1b: The environment is often less accessible to individuals with SCI/D and increases fall risk |
| (i) "I'm pretty good about using my wheelchair and going up slopes. . . my fear is more going down. . . that's where I fear that if I lose control, I am going to fall and I do not want to fall."—F56, wheelchair user |
| (ii) "You know, most of us. . .before I had the operation. . .I had taken all these things for granted because I was fairly normal in my walking. And, so it wasn't a problem. So, I think it's just, yeah, letting people know and having some kind of, I'm not going to say a campaign, but. . ."—F20 |
| Category 1c: Strategies to decrease risk of falling are individualized |
| (i) "Go down the stairs very carefully. I always hold the handrail, always. Now, walking, well you can't see everything, but I do watch for ice."—F74, ambulator |
| (ii) "And, while in [the bathtub], like standing up, there was nothing really to hold onto, so there was a risk of falling and to decrease most of that, we got the chair."—F40 |
| **Theme 2: Influences on the individual's perception of their fall risk** |
| Category 2a: Family members, friends, and pre-SCI/D falls |
| (i) "Yeah, right now because I actually have fallen, I have tripped before I have the spinal cord injury, so. . .about this is hoping that I can learn things"—F11 |
| (ii) "I already had a friend fall 30 feet, and he's dead."–F74 |
| Category 2b: Falls experienced since SCI/D and realization of consequences |
| (i) "Yeah, once I remember, somebody did grab me one time. When I fell, I dropped my cell phone and I went down, smacked my hands, and this guy grabbed me from behind."—F30, ambulator |
| (ii) "impact was like the first two falls, I was pretty cautious, I was actually worried, I actually wasn't really sure what to do or how to go about it. The rest of [the falls]. . .just hurt, but it didn't really do. . .any damage."—F40 |
| **Theme 3: Experiencing life differently due to increased fall risk** |
| Category 3a: Ability to participate in meaningful activities |
| (i) "I'm more cautious, I'm more observant, I guess I avoid what I consider to be dangerous places or conditions,"—F20 |
| (ii) "Ultimately, it [presence of accessible parking] helps decide where I shop, where I go for my recreation, you know,"—F20 |
| (iii) "I can't get to do the things that I really love to do or I did before"—F44 |
| (iv) "When the average person walks, they're not thinking about walking, you know, they're looking about, they're talking, they're thinking of other things. When I walk, I have to constantly be looking down"—F20 |
| Category 3b: Psychological impact |
| (i) "I'm always conscious and scared of almost everything I do."—F44 |
| (ii) "I do find it limiting, my life experiences, I find that it gives me attention or stress that I don't think is healthy, when every day you come across situations or places where suddenly your stress level rises because you're afraid of falling."—F20 |
| (iii) "there's a whole social aspect of my life that's been snatched away as well because of all of this. My confidence level. . . it's changed a lot."—F30 |
| **Theme 4: Falls training in rehabilitation can be improved** |
| Category 4a: Varying opinions/experiences |
| *Subcategory i. Varying experiences of falls training in rehabilitation* |
| (i) "They worked on improving my ability to walk, they worked on improving my balance, they worked on strengthening my feet and ankle and lower leg muscles, but nothing related to falls."—F20 |
| (ii) "I had a bit of training with one of my physios. . .when I was here though, it was really limited, like I left with a walker, so falls training, I didn't do a whole lot of it here. Most of what I learned, it's been out of here"—F30 |
| (iii) "Big focus [on falls training]. For one, we had a couple studies, but after every visit from home, like I would get asked these questions, like was there something that potentially could have led to a fall or was there something that led to a fall? So, those were things that were always asked and worked on every week."–F40 |
| *Subcategory ii. Varying opinions about the adequacy of falls training in rehabilitation* |

*(Continued)*

**Table 2.** (Continued)

| |
|---|
| (i) "I would say it is adequate right now because I've never fallen, I've been released for a year and I've never fallen."—F11 |
| (ii) "Oh no, I wish I had it. Yeah, I think it's really vital, I think it's very important because you don't realize. . .it's like setting somebody out there on their own and suddenly, they experience all of these dangers which were not dangers before. And, so you're not expecting them."—F20 |
| *Subcategory iii. Varying opinions about when falls training should be provided* |
| (i) "In a place like [deidentified rehab center], it could be done or should be done before the person is released, certainly and probably early on."—F20 |
| (ii) "So, maybe it could be a separate follow-up to people who have taken physiotherapy beyond the stage they're taking out patient, beyond the outpatient stage."—F11 |
| Category 4b: Recommendations for fall prevention programs |
| *Subcategory i. What falls training should include* |
| (i) "Well, I think the basic techniques, how to prevent falls."—F11 |
| (ii) "I think ideally it would include some practice falling like on a mat."—F09 |
| *Subcategory ii. Falls training delivery: Who and how* |
| (i) "Honestly, I would say physio. . .'cause they're quite possibly the most active person that you have in rehab, but then they deal with pain and other similar issues, like how to help you keep your balance, how to help you not fall, how to get up from a fall."–F40 |
| (ii) "Occupational therapists have an understanding, particularly of the upper body, but I think the physiotherapist would have probably have a better understanding of overall muscles. What you can do. Which ones you can use. So, sort of overall they know overall, I'd say, the best."–F09 |
| *Subcategory iii. Falls training should continue past outpatient rehabilitation* |
| (i) "You know what would've been helpful is actually if I did have somebody called me up a month after and says, 'okay, come in, we're going to do some fall training exercise.' That would've been helpful."—F30 |
| (ii) "I have a lot of concerns about [falls] when I'm aging. There's a lot of stuff, because I don't know how, yeah, I just don't know how things are going to work."–F56 (wheelchair user) |

environmental hazards were described by participants as "unsafe," "treacherous," "dangerous," and a "big concern." F09 described the difficulty of using a walker outdoors in winter weather: "With snow you are never sure if there is any ice underneath it. It's just uneven and it's unsafe walking." Several participants suggested that certain environmental hazards such as stairs were compounded by the added risk of weather and surface conditions: "For someone like me, who has a cane in one hand, it can also be treacherous in the winter or when it's raining out, if the steps are wet, if there's garbage on the steps" (F20). Participants explained that many of these issues were a result of decreased accessibility for individuals with physical disabilities living in the community, including long staircases (without an accessible alternative such as an elevator), decorative non-functional railings on staircases, uneven floor surface transitions (e.g. carpets in the entrance to a store), public washrooms without proper doors or handrails for wheelchair users, and a lack of accessible parking spots close to buildings (see Fig 2). For example, F30 explained a photo they took of an inaccessible public transit station:

> It's just a huge worry, big concern. . .and like some of the [deidentified public transit] places have elevators and some don't, so [deidentified station] where my parents live, when I go there every Sunday night for dinner, it's tough cause there's no elevator or escalator or anything. And, so that's a long stairway and it's a concern.

Many of these issues were attributed, in part, to a lack of understanding of fall risks from community members. As a result, the environment was not always constructed or maintained properly to allow accessibility to those with mobility limitations, increasing their risk of falling.

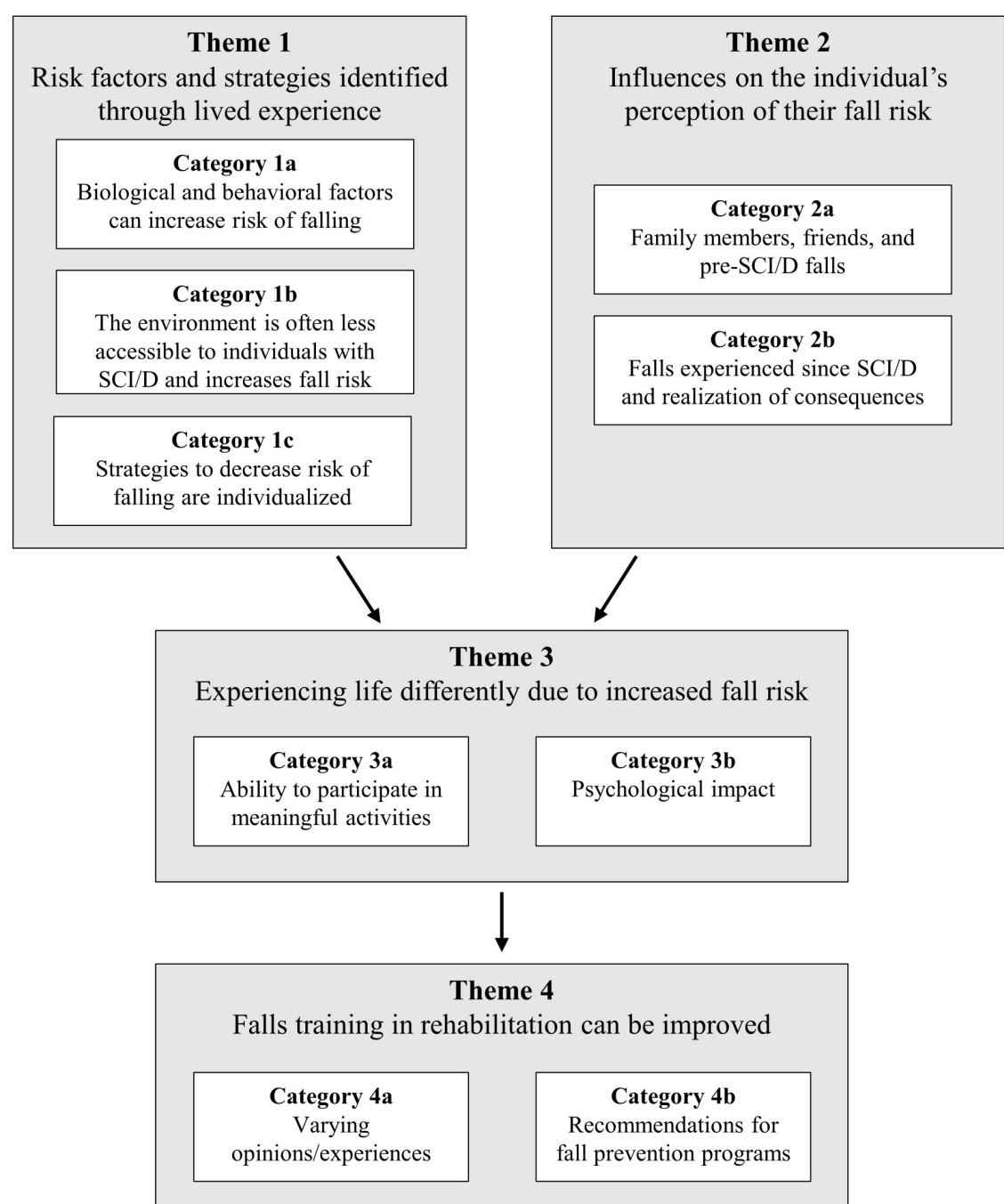

**Fig 1. Themes and categories.** Four main themes concerning falls and fall risk during the subacute phase of SCI/D are shown along with the categories within each theme. Risk factors and strategies identified through lived experience (Theme 1) and Influences on the individual's perception of their fall risk (Theme 2) lead to Experiencing life differently due to increased fall risk (Theme 3). Based on these lived experiences, participants felt that Falls training in rehabilitation can be improved (Theme 4).

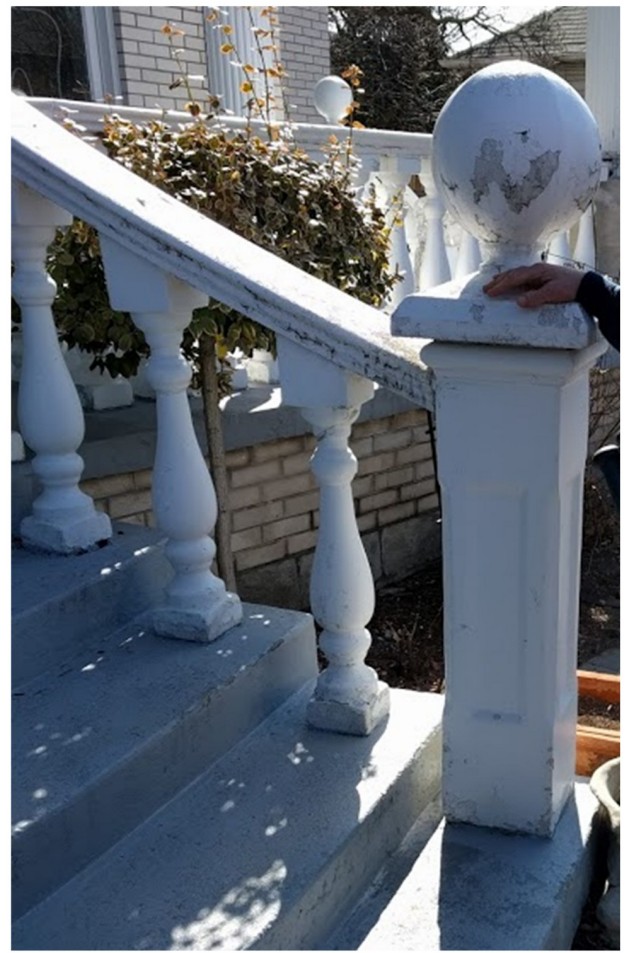

"What I see is…a railing that's not very functional…I actually went to this place earlier this week by myself. I drove over there, got my walker…once I got to the stairs I called them from outside. I called the office and could you send somebody to help me up these stairs? It just sometimes people don't realize that not every railing is equal." (F09)

**Fig 2. Inaccessible environments.** Photo taken by F09 highlighting how the community environment is often inaccessible and increases fall risk.

"Sometimes people don't realize that not every railing is equal," explained F09, referring to railings on the staircases of private businesses that are not functional or helpful to someone who requires them for safe mobility. F20 suggested, "There needs to be a smooth sidewalk policy. I mean you never realize it. As a normal walker, I don't think people realize how bad the sidewalks are and how they can be very dangerous to people like me." It was acknowledged by F74 that not all of these environmental factors have the ability to be changed or improved, and that it was inevitable that these situations would occur:

> There's no way you're going to avoid all the hazards, it's just impossible. So, is there any way to really improve it? There's no magic shoes that are going to, 'Oh there's something coming up, let's jump this guy over it', it's not going to happen. I can't think of any way to improve it.

**Category 1c: Strategies to decrease risk of falling are individualized.** One strategy that was consistent across the majority of participants was paying close attention to their surroundings when ambulating or using the stairs. F30 explained, "You don't think about walking and now it's something that I have to think about 'cause it doesn't come naturally. It's not a natural thing anymore." Strategies were creative and specific to the individual, such as one participant marking the edges of the stairs in their vacation home with red tape so that they could see the transitions between steps and reduce their risk of falling on the stairs: "I actually have to mark the stairs myself" (F20) (see Fig 3). Other participants also shared modifications made to their homes to decrease their fall risk. For example, F40 explained, "My cousin actually put a couch at the bottom of the step," so that he had somewhere to rest because he had previously had a fall at the bottom of the stairs due to fatigue. There were strategies that involved the need to "plan ahead" to reduce the risk of falling such as needing to "walk around and loosen up" (F30) before attempting the stairs. Participant F56 (wheelchair user) asked about accessibility before attending an event: "I went to a party on Saturday night and I specifically asked the host, just 'cause it was a party room, 'what's the bathroom like?' And, so she was kind enough to send me a picture and so it worked fine."

Several participants discussed the importance of advocating in the community as a way to address the factors that increase their risk of falls. For example, F20 explained the following approach to dealing with uneven pavement in the community:

> If there's really uneven pavement or. . .where it sticks up. . .quite a bit and can make someone fall or if there's unevenness or cracked pavement or where let's say a tree is planted and the roots are causing the sidewalk to swell, I phone [the city], but I believe I shouldn't have to do that.

A similar experience was shared by F56 (wheelchair user), who encountered public washrooms with broken handrails and doors to stalls that prevented wheelchair access: "If I come across it, I'm just going to write to whoever and say. . .by the way this is what I found, and it makes it a little more difficult for me to use your washrooms."

## Theme 2: Influences on the individual's perception of their fall risk

Participants discussed factors that influenced their personal perception of their fall risk; the fall experiences of their friends and family, pre-SCI/D falls and falls they had experienced since sustaining SCI/D.

**Category 2a: Family members, friends, and pre-SCI/D falls.** Several participants discussed friends or family members (both with and without SCI/D) who had experienced a fall

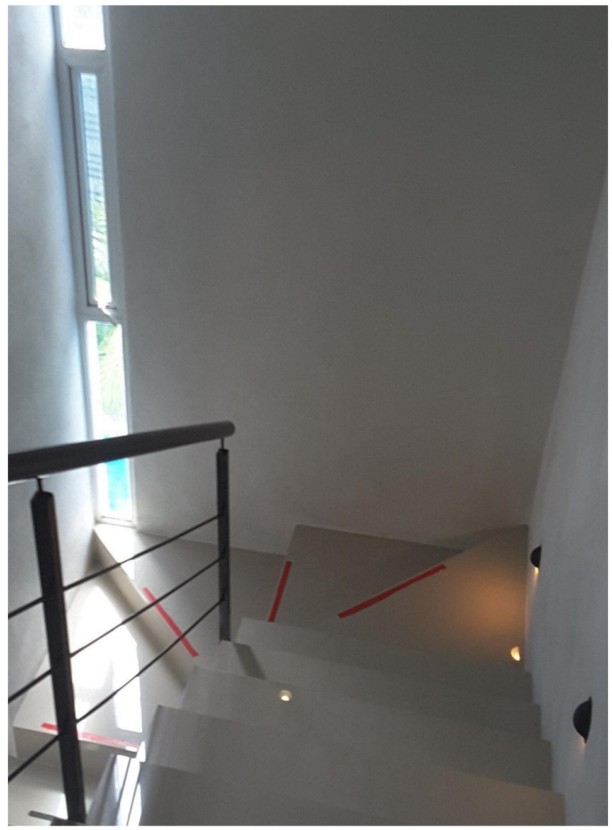

"...[vacation home] is one of the places that I go to quite often, and I actually have to mark the stairs myself... I just use the red adhesive tape so that pulls right back up. So I don't leave it their when I leave.... I try to avoid places where there are a great number of stairs or if they're curved. But, you can't always do that, so when I have to use stairs, such as that, I, first of all, as soon as I see them I feel uptight, okay. I feel stressed. But, then I really have to slow down the whole process of going up or down stairs. I really have to think about what I'm doing. I have to hang on tighter to the railing. I have to make sure I put my cane down extra properly." (F20)

**Fig 3. Strategy to reduce fall risk.** Participant F20 describes a strategy used to decrease fall risk when descending stairs in a vacation home.

that led to serious consequences. F20 stated the following, relating a family member's experience to his own situation: "I know that my sister took a terrible fall in a major department store because of the change in height [of the floor surface]. And, she also had the same operation as I did." Similarly, F56 recalled a fall she witnessed and voiced her concerns about what she would do if she experienced a fall while in her wheelchair:

*Cause I've seen it, I was in [deidentified] when a guy missed the curb and he fell over in his wheelchair, and I'm like, this is before this happened to me, you know a lot of people were out there trying to help him, so but yeah, I wouldn't know what to do.*

F11 recalled what happened when his elderly mother (who was living alone at the time) experienced a fall in her home. He shared that this fall changed her life, emphasizing his concerns about the potential devastating consequences a fall could have on a person's life:"With my experience of my mother...you see her life completely change...when she had a fall. Right, and that you remember. Like she was reasonably healthy...you know basically functional, and all of a sudden she's a different person."

**Category 2b: Falls experienced since SCI/D and realization of consequences.** Four out of eight participants reported experiencing at least one fall since their SCI/D. All of the

reported falls occurred once participants were discharged from inpatient rehabilitation, except for one fall, reported by F40, that occurred on a weekend pass home during his inpatient rehabilitation stay. Participants discussed how these falls had affected them. F40 explained, "[A fall is] very painful. . .extremely painful. . .wouldn't want to have to go through that again." F56 (wheelchair user) described a time when she was unable to get back up due to a fall that caused an injury: "I had hurt my wrist from the fall, I thought, 'Well, this isn't going to work.' So, I got on my back and my husband dragged me to our bedroom and then picked me up and put me on the bed." F09 described a time he fell and his wife "had to get two neighbours to help get [him] up." For both of these participants, they required assistance from others, compromising their independence. These experiences had an influence on how the participants viewed falls, the potential consequences of a fall, and heightened how fearful they were of falling.

## Theme 3: Experiencing life differently due to increased fall risk

Throughout all of the participants' interviews, it was clear that their increased risk of falling and their strategies to mitigate risk had a significant impact on their lives. Participants avoided situations that would increase their risk of falling and altered their participation in daily activities. These changes resulted in participants experiencing decreased independence and quality of life, and increased negative emotions.

**Category 3a: Falls/fall risk impacts ability to participate in meaningful activities and decreases quality of life.** Many participants explained how their fall risk impacted their ability to participate in meaningful activities. This could include having to alter the way they participated, dependence on others, or avoiding activities altogether because of the risk. Some participants were unable, "to do the things that [they] really love to do," (F44) and others were unable to go back to work: "Basically they tell me that 'til I could walk on my own and lift up 50 lbs, it would be ill-advised for me to come back," (F40). The need to focus on safety constantly to avoid a fall took away from the enjoyment of everyday activities. For example, "something as simple as going upstairs and carrying a glass of water," (F20) was a very difficult task that increased fall risk for someone who required a gait aid. The need to consider fall risk when making plans also had an impact on daily life for F09: "It makes it difficult to plan ahead. If there is snow overnight, then I would have to cancel my plans for the next day." F20 described how the need to focus on his fall risk took away from the experience during a vacation (see Fig 4). Similarly, F30 shared how a fear of falling had altered his participation in yoga, a leisure activity he used to love:

*"Yeah, I used to do a lot of yoga and now I have a real hard time with it, like I do a real beginner's class now. And, I can't do hardly anything, like I can't do a downward dog. I'm just actually getting back to doing sort of a kneeling thing, but for me to go down and up, it's really hard."*

Another ambulatory participant (F30) explained the effect of not being able to participate in activities that provided him with social opportunities such as yoga and cycling: "There's a whole social aspect of my life that's been snatched away as well because of all of this." F09 explained how his increased fall risk had limited his independence:

*Unless I got extra help, I wouldn't be mobile, I would be staying in waiting to see what can of soup is in the cupboard or to see what's on Netflix. I'd be calling the appointments and saying I can't make it.*

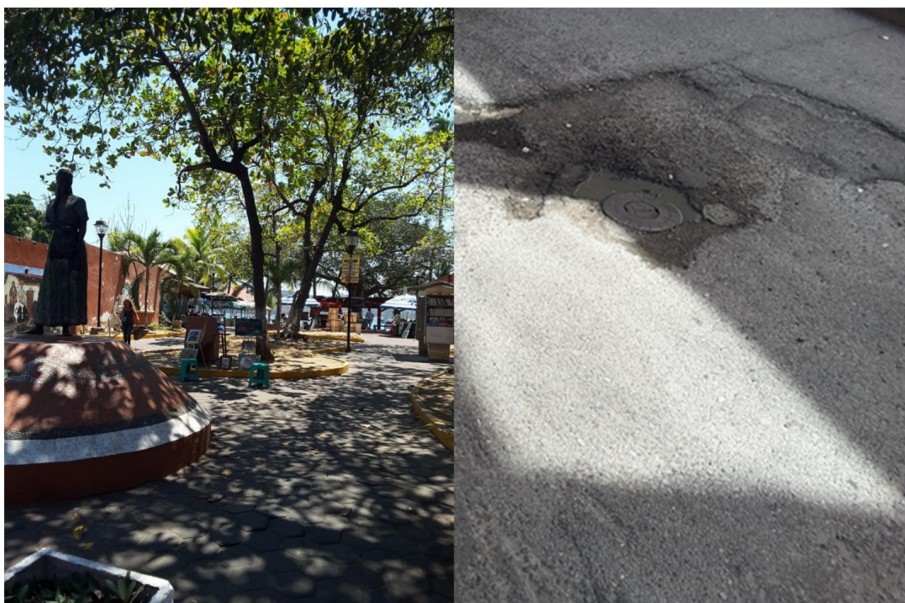

"I was in [deidentified city], [deidentified country] for a holiday and the buildings, the architecture, everything was phenomenal except that I couldn't see it because when I was walking, I had to be looking down all the time because their sidewalks and roadways were in such disrepair. And, so it really took away from the aesthetic for me, it took away my enjoyment." (F20)

**Fig 4. Impact on participation.** Photos taken by F20 illustrating how falls and fall risk impact participation in meaningful activities.

**Category 3b: Psychological impact.** Participants described increased fear, anxiety, worry, stress, concern, loneliness, embarrassment, and decreased confidence as consequences of their constant awareness of their risk of falling. F20 explained that, "something simple such as walking becomes stressful," for individuals who were ambulatory and at an increased risk of falling. F20 also described how this constant diligence about their fall risk became tiring: "This always being cautious and thinking about what you're doing. . .not only is it stressful, but it's also tiring." Another source of fear expressed by the same participant was regarding the possible consequences of a fall: "I'm terrified of falling down stairs because. . .I have [hardware] in my neck and so I'm really afraid that if I fall that I could kill myself or there would be major repercussions." One participant (F30) discussed the how living with an SCI/D have affected him: ". . . it affects me, like emotionally, I think about what it used to be like. . .you could say I feel ripped off or something like that, I just feel like, I guess there's a big hole." Several participants expressed concerns about what their future would look like and how that would affect their risk of falling. F74 was concerned about how much physical ability he would recover and how that would affect his ability to prevent a fall. He shared that he had a ladder set up to work on the eaves trough and hasn't moved it because it is unsafe for him to complete this task with his current physical limitations: "How far will I regain?. . .if I had to reach out. . .I let go of one of

the [ladder] rungs and I'm doing something and I start to fall back, I couldn't reach for the rung, I don't have the range." F56 (wheelchair user) experienced a fall on the day she arrived home from inpatient rehabilitation which resulted in an injured wrist. This participant was worried about how another fall would affect her physical progress: "Just the fear of falling. . .I am just getting back into being able to progress again and I really do not need something to put me back again." Both of these individuals were concerned that their ability to continue to improve physically would be affected and as a result, they would continue to be limited in their daily life due to a risk of falling.

### Theme 4: Falls training in rehabilitation can be improved

Something that was clear across all of the interviews was that the approach to falls training in SCI rehabilitation could be improved. Many participants offered suggestions and appeared to be in agreement that falls training should be comprehensive and individualized. Two categories with subsequent subcategories arose from this theme.

**Category 4a: Varying opinions/experiences.** This category addressed the various differences in opinions that were expressed throughout the interviews, which reflects the need for falls training to be specific to the individual and their needs.

*Subcategory i*: *Varying experiences of falls training in rehabilitation*. Several participants reported receiving some type of falls-related training during rehabilitation; however, there were also participants that stated they did not receive any falls training whatsoever. Those who reported receiving falls training typically described their physical therapist teaching them how to get up from a fall: "Inpatient [rehabilitation] was basically training my caregiver on how to safely get myself up. The training in outpatient [rehabilitation] was how to get up independently with minimal assistance" (F09). Several participants agreed that falls training was "not as much of a goal as mobility," and that the focus of rehabilitation was more about regaining strength and getting the person "upright" rather than on falls prevention. F11 suggested that if there was any falls training provided, he had likely forgotten about it, because mobility was more important to him at the time: "I only remember the things that were important to me. . .'cause. . . I want to get myself to the stage where I can walk." One participant (F11) who stated that he did not receive any falls training explained that the reason for this was because he was using a walker for ambulation: "No, I did not get any because. . .at that time, I can only walk with the walker." Another participant (F74) felt that he only received a minimal amount of falls training because he was ambulatory: "I think they didn't focus on it 'cause I could walk."

*Subcategory ii. Varying opinions about the adequacy of falls training in rehabilitation*. Some participants were satisfied with what they had received in rehabilitation, but others felt they did not receive adequate falls training. F56 (wheelchair user) stated that she did not receive any training about falls when using a wheelchair: "I think [the falls training] was adequate for falling from a mobility device. . .I don't think I'd know what to do if my chair tipped over." One participant (F30) who reported he did not receive falls training in outpatient rehabilitation voiced his concern: "I think from a safety standpoint of view. . .it would've been something that should've been there, maybe." One participant who was satisfied with their experience shared, "It was very helpful because it gave me both the knowledge for how to get up safely, but it also gave me more confidence" (F09).

*Subcategory iii. Varying opinions about when falls training should be provided*. There were also different opinions about when falls training should be provided throughout the rehabilitation process. Some of the opinions included: "towards the end of the inpatient or the outpatient, nearing discharge," "beyond the outpatient stage," and "as soon as you're mobile." Some

participants believed it was important to provide falls training before discharge from inpatient rehabilitation because, "the fall could happen at any time." In contrast, F09 discussed why falls training should be provided during outpatient rehabilitation:

> *For me, the first time I did it [falls training] as an inpatient, it was tough to do because I didn't realize I didn't have that independence. But, when I did [it] in my outpatient. . .it was a great confidence builder, not only my confidence but my caregiver's confidence.*

One participant (F74) discussed how overwhelming inpatient rehabilitation could be, and that falls training should not be provided too early on:

> *I think once you start into physio and whatever your program's going to be, I think you need that time to adjust to this place first, don't throw somebody off the cliff right away, so give them a time to assimilate into this environment, then do it.*

**Category 4b: Recommendations for fall prevention programs.** Participants had many suggestions and ideas about what an ideal falls training program should include, how it should be delivered, and the importance of ongoing falls training.

*Subcategory i. What falls training should include.* Participants discussed the importance of educating about fall prevention, for example: "Some kind of an in-service. . .or presentation. . .saying, okay, now that you're leaving here. . .there are going to be some real dangers out there that perhaps weren't dangers to you before," (F20). Learning how to fall safely was suggested by several participants and F11 stated, "The important thing is, you know how to make sure that if you do fall. . .you don't hurt yourself badly." One suggestion from F20 was about learning how to use a mobility device to break the fall, something that he learned through experience:

> *I wish I had been taught at [deidentified rehab centre]. . .that the millisecond that you realize that you're going to fall that you can use your cane to help break the fall. . .wrapping yourself around the cane. . .and using it as a support. So, yeah, that would've helped, I mean I had to discover that myself and I think it's something simple that could've been done.*

One participant (F09) recognized the importance of training a caregiver about how to help the person up from a fall: "To safely train a caregiver to be there themselves or helping a caregiver basically guide. . .other people." Several participants also suggested learning how to check for injuries in the case of a fall: "Make sure that you didn't hurt yourself in any way. And, if you do feel like you're hurt in some way, go to the ER," (F40). Two participants suggested customized resources that were made specifically for the individual:

> *I don't think that handouts would be that effective. It might be. . .if they basically assessed you on how well you do. . .if they could say just as a reminder, this is what you did the last time, you rolled over on the left side, you then pushed up, for me it was 'roll over on the left side, take my left arm underneath, push up with that, so that your weight is on your left arm, and then you rotate to shift your weight onto your right arm, and then you know. . .a step by step', so that could be helpful guide. (F09)*

*Subcategory ii. Falls training delivery: Who and how.* Majority of participants were in agreement that a physiotherapist would be the best person to deliver falls training sessions. F09 explains:

*They [physiotherapist] know the overall strength and limitations I physically have. I think they know just some tools you could get, like how to use a walker to get up, how to use your wheelchair if you fall out of a wheelchair, what are the things you should and shouldn't do.*

However, F20 also acknowledged that an "interdisciplinary approach" would be ideal: "For instance, I don't think a physiotherapist would be the only person who could deliver a seminar or a talk or a conversation [about] falling." Several participants were interested in the idea of group classes for individuals with SCI: "I suppose a group thing's not bad because you'd have different points of view and stuff like that," (F30). F20 shared the following opinion about falls training in rehabilitation:

*It's one thing to give lip service to something, it's another thing to actually do it and make sure that it's done and the way to do that, I found, is to make sure that it's part of the structure, that it's an integral part of the structure of the program, okay.*

*Subcategory iii. Falls training should continue past outpatient rehabilitation.* One idea that was suggested by participants was that falls training should continue past the stage of active rehabilitation. Participants explained that "peoples' abilities are always changing, [their] situations are changing," (F20), whether due to age or other life events, and so falls training should reflect that. Two participants even discussed how the follow-ups from the current research study helped them to reflect on their own fall risk, which suggested some type of falls training follow-up would be beneficial. F30 (ambulator) explained, "Being part of this study for such a long time now has really helped. . .having somebody call me every month was a big help 'cause it got me actually thinking, right? And, so I was more self-aware with that little bit."

## Discussion

This qualitative study was among the first to explore the perceived impact of falls and the risk of falling among individuals with subacute SCI/D during the transition from inpatient rehabilitation to community living. Four themes were identified from the interview and visual data. First, risk factors and mitigation strategies were identified through lived experience. Participants identified biological, behavioral and environmental risk factors, with environmental factors being noted as difficult to modify. Second, participants described how experiences with falls, whether experienced by themselves pre- or post-SCI/D or experienced by friends or family, influenced their perception of fall risk. Third, participants described experiencing life differently due to their increased fall risk. More specifically, their participation in meaningful activities was decreased, which meant reduced social engagement and psychological impact. Fourth, participants suggested that falls training in rehabilitation could be improved. Although their experiences with falls training during inpatient and outpatient rehabilitation were varied, participants agreed that falls training should be comprehensive, individualized and should extend beyond outpatient rehabilitation as each person's abilities and circumstances change over time. A synthesis of the findings resulted in recommendations for fall prevention education and training throughout the first year after SCI/D (see Table 3). These recommendations, which were derived from the participants' data and prior literature, are applicable to and desired by clinicians and rehabilitation programs [18, 19] involved in facilitating the transition to community living after SCI/D.

All eight participants in this study experienced non-traumatic damage to their spinal cord. To date the majority of the research on falls in this population has focused on those with traumatic causes of injury [2, 3, 5, 8–10, 12, 13, 37]; however, the demographics of SCI/D are changing [38–41]. The incidence of non-traumatic causes of SCI/D is increasing, with the

**Table 3. Recommendations for fall prevention education and training in the first year post-SCI/D.**

| Recommendations: |
| --- |
| • Provide real-world examples of the causes and consequences of falls after SCI/D through peer involvement |
| • Provide a safe environment to experience controlled falls and near-falls under the supervision of a physiotherapist |
| • Provide training in advocacy skills, possibly with peer involvement |
| • Provide fall prevention training education and training in the context of each mobility device used by an individual (e.g. wheelchair and walker in the case of a part-time ambulator) |
| • Continually re-visit fall risk and fall prevention education and training throughout the first year post-SCI/D |
| • Include caregivers in fall prevention education and training (e.g how to assist following a fall, how to check for injuries) |

SCI/D, spinal cord injury or disease.

prevalence now estimated to exceed that of traumatic injury in some countries [42, 43]. In Canada, it is estimated that there are two times as many non-traumatic cases of SCI/D each year compared to traumatic injuries [41]. Consistent with the increase in non-traumatic SCI/D, the age of onset of SCI/D is increasing and more individuals are presenting with pre-existing health conditions [38, 41]. Both increasing age and co-morbid conditions are known risk factors for falls [44]. All but one of our participants were aged ≥50 years, and half of the participants were aged 65 years or greater. However, these ages are typical for SCI/D rehabilitation in Canada. The majority of those with SCI/D receiving inpatient rehabilitation in Canadian hospitals are >50 years of age [41], with the mean age of those receiving inpatient rehabilitation due to non-traumatic injuries being 59.0 ± 18.5 years [45]. The issues and concerns described in this study may reflect both aging and sustaining non-traumatic SCI/D, and may have differed if our participants were primarily younger individuals with SCI/D or those with traumatic injuries.

The participants in this study learned about their individualized fall risk and fall prevention strategies through experience with falls and near-falls, which is consistent with reports from the chronic SCI/D population [2, 11]. The current findings suggest this experiential learning strategy is adopted within the first year post-SCI/D. However, in contrast to the chronic population, those in the subacute phase of SCI/D referenced learning through the experiences of friends and family in addition to their own experiences. This difference may reflect that those in the subacute phase of injury have had fewer exposures to falls, and thus, less experience to draw from. There are several implications of this finding. First, fall prevention education during the subacute phase may benefit from the inclusion of real-world examples of falls, perhaps from those living with chronic SCI/D. Peers have been previously identified as a valuable and credible source of information concerning falls [13]. The inclusion of peers may increase the saliency of fall prevention education and training during this early phase post-SCI/D, as several participants in the current study indicated that fall prevention was not a priority for them during inpatient and outpatient rehabilitation. Second, providing a safe environment in which to experience losses of balance when standing, walking, wheeling and/or transferring may provide opportunities for experiential learning. While reactive balance training, which provides repetitive exposure to falls, has been studied in the chronic, ambulatory SCI/D population to date [46], there may be value in investigating its feasibility and efficacy in the subacute population and modifying the training to accommodate wheelchair users.

Participants in this study described encountering inaccessible environments that contributed to a greater fall risk, which is consistent with prior work studying fall risk in individuals with chronic injury [2, 3, 6, 9, 11]. Following discharge from the hospital, individuals with newly acquired SCI/D find they are "living in a changed world" [47] in that their interactions

with the environment are different compared to that experienced pre-injury. Consistent with other studies [2, 11], the current participants mentioned making changes within their indoor living environment to reduce their fall risk and establish "an accessible proximal environment" [47]. However, the current participants with subacute SCI/D noted the challenge of changing community environments. In contrast, those living with chronic SCI/D have explained that over time they learned strategies to influence the people and communities around them, such as educating others about accessibility [47]. Individuals with chronic SCI/D have previously identified advocacy skills as important for fall prevention; for example, having the confidence and knowledge of how to advocate for the removal of fall hazards [2, 13]. Hence, advocacy skills should be included in fall prevention education and training.

Participants provided numerous suggestions for how fall prevention education and training could be improved. They agreed that the initiatives should be individualized; a suggestion that has been previously expressed by the chronic SCI/D population [13] and is also recommended in the Canadian Stroke Best Practice Guidelines [48]. Participants in the current study, along with those living with more chronic SCI/D [13], felt that fall prevention education and training should be available beyond inpatient and outpatient rehabilitation, as mobility status and life circumstances change over time. Similarly, the Canadian Stroke Best Practice Guidelines recognize the need for ongoing fall risk screening, suggesting it should be performed at admission to rehabilitation, at times of transition, after each fall and when one's health status changes [48]. Mechanisms for offering fall prevention education and training beyond outpatient SCI/D rehabilitation may be an important topic for future research and development.

Participants did not agree on when fall prevention education and training should be first introduced following SCI/D. Some participants felt fall prevention lacked relevancy during inpatient rehabilitation. Similarly, physical and occupational therapists and hospital administrators have questioned whether inpatient rehabilitation is the most appropriate setting for fall prevention education and training as the circumstances of each individual's fall risk in the home and community are unknown at this stage [18, 19]. Moreover, individuals with SCI/D have previously described hospitalization after SCI/D as a time of distress and confusion, and were unprepared to receive education [49]. Several of our study participants shared similar sentiments; inpatient rehabilitation was a time when they felt overwhelmed and did not appreciate the relevance of fall prevention education and training. The optimal timing of fall prevention education and training warrants future study.

The current study has several limitations to note. First, the majority of interviews were conducted in person; however, three interviews were completed over the phone. In addition to lacking non-verbal communication (e.g. body language, hand gestures, gaze), the dialogue of phone interviews is known to differ from that of in person interviews [50]. For example, interviewees tend to ask for more clarification during phone interviews and phone interviews tend to be shorter in duration [50]. Second, participants were asked about their experiences with fall prevention training; however, a recent study indicated that physical and occupational therapists implement fall prevention initiatives during inpatient and outpatient rehabilitation without explicitly labeling these initiatives as fall prevention training [19]. Hence, the participants' reflections in the current study may not reflect the entirety of the fall prevention training received. Third, although our sample size of eight participants was within the recommended range for photo-elicitation research [11, 28–30], this sample size is small. Data saturation, or the point when no changes or additions to the codebook result from additional interviews [51], may not have been reached in our study. However, prior research that also involved a nonprobabilistic, purposive sample demonstrated that 80% of the codes were identified after the first six transcripts and 92% of the codes after 12 transcripts [51]. Hence it is reasonable to assume that additional interviews would not have led to significant changes to the codebook.

Fourth, we were surprised to recruit only eight participants from the 45 individuals sent the recruitment flyer. We did not systematically track the reasons why participants chose not to participate, as potential participants contacted the research team only if they were interested in participating in the photo-elicitation study. However, a few participants volunteered reasons why they chose not to participate, which included being too busy with community-based rehabilitation or other medical appointments, and preparing for a return to work. Fifth, all eight participants had experienced non-traumatic SCI/D; hence, the perspectives of those with subacute, traumatic injuries are not reflected in the study findings. Lastly, all but one participant identified as male (i.e. 87.7% of the sample). In Canada, approximately 57% individuals with non-traumatic SCI/D are male [45], meaning that males were over-represented in our sample.

In conclusion, the study findings have described the impact of falls and fall risk during the transition from hospital to community living from the perspectives of eight individuals with non-traumatic SCI/D. Participants discovered their fall risk factors and fall prevention strategies through lived experience with falls, which included falls experienced by themselves as well as friends and family. A high fall risk caused participants to alter participation in daily activities, increased negative emotions and reduced independence and quality of life. Participants' experiences with falls and fall prevention varied; however, all agreed that comprehensive and individualized fall prevention education and training was needed.

## Supporting information

**S1 Checklist. COREQ checklist.** Completed COREQ (Consolidated criteria for reporting qualitative research) checklist.
(DOCX)

**S1 File. Interview guide.** Semi-structured interview guide.
(DOCX)

## Author Contributions

**Conceptualization:** Kristin E. Musselman.

**Data curation:** Olinda D. Habib Perez, Katherine Chan, Kristin E. Musselman.

**Formal analysis:** Olinda D. Habib Perez, Samantha Martin, Katherine Chan, Kristin E. Musselman.

**Funding acquisition:** Kristin E. Musselman.

**Investigation:** Olinda D. Habib Perez, Kristin E. Musselman.

**Methodology:** Olinda D. Habib Perez, Samantha Martin, Hardeep Singh, Karen K. Yoshida, Kristin E. Musselman.

**Project administration:** Olinda D. Habib Perez, Katherine Chan, Kristin E. Musselman.

**Resources:** Hardeep Singh, Karen K. Yoshida, Kristin E. Musselman.

**Supervision:** Kristin E. Musselman.

**Visualization:** Kristin E. Musselman.

**Writing – original draft:** Olinda D. Habib Perez, Samantha Martin, Kristin E. Musselman.

**Writing – review & editing:** Olinda D. Habib Perez, Samantha Martin, Katherine Chan, Hardeep Singh, Karen K. Yoshida, Kristin E. Musselman.

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
