## [Decision Letter · Decision Letter 0]

28 Feb 2022

PONE-D-21-15719A qualitative photo-elicitation study exploring the impact of falls and fall risk on individuals with sub-acute spinal cord injuryPLOS ONE

 Dear Dr. Musselman,

Thank you for submitting your manuscript to PLOS ONE. Two experts in the field have reviewed your manuscript. While both reviewers found the manuscript to make a notable contribution to our understanding of falls and fall risk in people with SCI, they each provided detailed feedback outlining required revisions prior to further consideration. Specifically, issues related to qualitative methodology (e.g., coding scheme, saturation, sampling), and how any methodological choices may have influenced study results (e.g., older sample may suggest aging related changes in fall risk) were highlighted by both reviewers. Please see each reviewer's detailed comments below, making sure to address all comments in your response. We invite you to submit a revised version of the manuscript that addresses the points raised during the review process.

We look forward to receiving your revised manuscript.

Kind regards,

Andrew Sawers, PhD

Academic Editor

PLOS ONE

Journal Requirements:

3. Your abstract cannot contain citations. Please only include citations in the body text of the manuscript, and ensure that they remain in ascending numerical order on first mention.

4. We note that Figures 2, 3 and 4 in your submission contain copyrighted images. All PLOS content is published under the Creative Commons Attribution License (CC BY 4.0), which means that the manuscript, images, and Supporting Information files will be freely available online, and any third party is permitted to access, download, copy, distribute, and use these materials in any way, even commercially, with proper attribution. For more information, see our copyright guidelines: http://journals.plos.org/plosone/s/licenses-and-copyright.

a. You may seek permission from the original copyright holder of Figure 2, 3 and 4 to publish the content specifically under the CC BY 4.0 license. 

5. Please upload a copy of Supporting Information "Consolidated criteria for reporting qualitative research (COREQ):

754 32-item checklist" which you refer to in your text on page 39.

Reviewers' comments:

Reviewer's Responses to Questions

**Comments to the Author**

1. Is the manuscript technically sound, and do the data support the conclusions?

Reviewer #1: Partly

Reviewer #2: Yes

2. Has the statistical analysis been performed appropriately and rigorously? 

Reviewer #1: N/A

Reviewer #2: Yes

3. Have the authors made all data underlying the findings in their manuscript fully available?

Reviewer #1: Yes

Reviewer #2: Yes

4. Is the manuscript presented in an intelligible fashion and written in standard English?

Reviewer #1: Yes

Reviewer #2: Yes

5. Review Comments to the Author

Reviewer #1: This manuscript addresses an important topic of falls in individuals with subacute SCI/D as they transition from inpatient rehabilitation to home. As most research has been conducted in chronic SCI, this adds important information to the body of literature on falls and SCI. Overall, the paper is well written. Suggestions for revisions are provided here.

Abstract- The abstract is well written summarizing key points of the manuscript.

Minor comments- Line 27 Instead of “inpatients with a new SCI/D…” consider “Individuals with SCI undergoing inpatient rehabilitation due to a new SCI/D…”

Introduction: The authors present a good rationale for investigating falls in the subacute population.

Line 68 – The authors state “only 7-13%...” The use of the word only implies bias that this percentage is not of concern. This number is still of concern and relevant to falls in the hospital setting. Suggest stating the percent of those in inpatient that fall and contrast that with a percent of individuals who fall after discharge so the comparison is seen more factually without the introduction of bias or minimizing the problem of falls in the inpatient setting.

Methods- Overall, the methods are described clearly.

Line 104, similar recommendation to what was said in the abstract on line 27

Line 114- The authors should provide additional information on why 7-10 was the target sample size. It says this was based on previous research - but what about the previous research led the researchers to arrive at this number? Was this enough to reach saturation of data for example? The COREQ checklist says saturation is addressed on p6-7 but this is the only line that addresses sample and saturation is not addressed directly.

Line 131-133 – It is great that the authors offered this modification though this does not seem to add anything to the manuscript

Line 153-155- recommend describing codes in the authors own words as it relates to this topic and citing reference 31 vs the direct quote as this may be more meaningful to the reader

Line 153-155 Authors generate ideas for codes and then SM applied the coding scheme. What happened between this step? How was the coding scheme agreed upon?

Line 157- Why is theme 4 the only one that has subcategories? Could the methods provide more information on the division of themes to categories to subcategories and how that was done?

Results-

171- The reader is left wondering why only 8 individuals participated when 45 were invited?

Lines 174 and Table 1 – The authors clearly establish that all participants were ambulatory except 1. This is clear and it seems unnecessary and distracting to then note “ambulator” after each participant code throughout the manuscript. The authors could indicate F56 as a manual wheelchair user if they want to distinguish, but since the reader already knows the rest are ambulatory, this feels redundant and distracting.

Line 189 and Table 2 – Nice summary of themes and supporting quotes to support the categorization. Category 2a only has 1 quote and 4b has no quotes. For consistency it seems each should have at least 2 quotes to show support for the categories.

The authors summarize each theme, category, and subcategory in the remainder of the results providing support from the interviews for each. It is unclear why some quotes are embedded directly in the paragraphs and some quotes are presented in italics. What is the difference in how these are presented and why?

Discussion-

Line 490 – Authors need to make clear that the recommendations came from the participant data

Lines 498-509 The authors address the issue that all of the participants had non-traumatic SCI and that half were over age 65. Did the authors accomplish their goal with this participant pool as the results could reflect SCI and age related fall risk factors? Results may have been different if the population was more representative of the demographics of SCI. The authors state in line 498 that “it is noteworthy”. This is more than just noteworthy and is a major limitation of the manuscript and should be addressed in limitations. Would the themes have come out the same way if younger individuals with traumatic injuries were included in the participant pool? The paper still presents good qualitative data for the 8 individuals with non traumatic SCI though this is a big limitation.

Bottom p28-top p29- Another concept that would be worth discussing related to the fall prevention training is individual’s readiness to learn. As stated in the limitations, PTs and OTs may include fall education and authors state that they may not label it as fall prevention training, but also patients may not be ready to hear or absorb the information at that time or think that falls would ever even be an issue given their perception of recovery and readiness to learn. The patient perception of it being included in their rehabilitation is important, regardless of if it was or was not, but acknowledging a possibility that it was included and not absorbed at the time is important.

Conclusion: Suggest revising the conclusion to directly connect to the themes as the conclusion does not fully support the data that is presented.

The conclusion in the discussion states that the study describes the impact of falls and fall risk. In the abstract the conclusion states that falls can have a significant impact on the first year of living… These are different and should be consistent, though see above suggestion for revision. ( It would also be helpful to know how many falls did occur during that 6 month period. Was this asked of the participants? While this is not qualitative data, it would help give some context) . One of the quotes specifically states the participant has not fallen, but beyond that the reader does not know.)

The conclusion also states that findings may inform future fall prevention initiatives. This is very generalizable. It may be more appropriate to state conclusions related to the themes and what was found with these 8 individuals. The variability of responses related to experiences and opinions about fall prevention appears to be a more clear conclusion from the data.

Reviewer #2: The goal of this qualitative study was to determine the impact of falls and fall risk among individuals in the sub-acute phase of SCI/D. Overall, the study is interesting and provides valuable information regarding subjective perception of this vulnerable population to better understand which situation increases/decreases fall-risk and its impact of their participation in walking activities. The findings from this study are clinically relevant to design customized fall prevention training programs available beyond inpatient and outpatient rehabilitation, as mobility status and life circumstances change over time, based on participants’ needs. While the manuscript is generally well written and fills the research gap, further clarifications are needed for certain sections.

Major comments:

Since the current demographics show that half of the total participants were above 65 years of age, did this subgroup of sample have a previous fall history or fear of falling associated with aging related changes even before the SCI/D? Do the authors consider that these individuals are more likely to restrict their participation in walking activities (due to aging related balance deficits, fear of falling) and thus might encounter fewer circumstances that might increase potential fall-risk?

Given the heterogenous sample including varying level of neurological involvement, could the author comment about the influence of perceived fall-risk for individuals with cervical vs lumbar levels. Also, did the authors come across any interaction between behavioral factors and level of involvement. For eg. Individuals with cervical level of involvement showed lower risk-taking behavior and are more cautious of their surroundings resulting in identification of more risk-factors and use of strategies to prevent falls, compared to those with lumbar level who might demonstrate greater risk-taking behavior.

Was there any sub-analysis performed to determine whether those who experienced falls during the 6 months after discharge vs non fallers demonstrated any difference in their perceived fall-risk or impact of falls in their activity participation? Additionally, during this period, did the participants undergo any balance and/or locomotor training or fall education intervention? Was this information collected and analyzed to understand the potential implications on change in fall-risk factors among those who received intervention vs those who did not?

Do authors think that information regarding participants housing is necessary to be included as a contributor towards fall-risk, since inability to fall-proofing the house might lead to greater fall-risk?

Inclusion of uneven number of males and females might be a potential limitation of the study since gender specific disparities (if any) would be overlooked given that majority of participants were males.

During the in-person interview, authors mention about non-verbal cues were given. What was the nature of these cues and could it possibly influence the results? If yes, how did the authors control for participants bias?

Page 8 line 153-155: It would be helpful to be more specific regarding how the analysis was performed using the codes. Also, detailed explanation regarding ‘information that can be assessed in a meaningful way regarding the phenomenon’ will be valuable.

Minor comments:

Page 6 line 61: correct to ‘reduced work productivity’

Page 27 line 536-537: “However, the participants with subacute SCI/D noted the challenge of influencing community environments.” This sentence is confusing and not very clear.

6. PLOS authors have the option to publish the peer review history of their article (what does this mean?). If published, this will include your full peer review and any attached files.

Reviewer #1: No

Reviewer #2: **Yes: **Shamali Dusane

---

## [Author Response · Author response to Decision Letter 0]

13 Apr 2022

Please see uploaded file labeled 'Response to Reviewers'.

---

## [Decision Letter · Decision Letter 1]

26 May 2022

A qualitative photo-elicitation study exploring the impact of falls and fall risk on individuals with subacute spinal cord injury

PONE-D-21-15719R1

Dear Dr. Musselman,

We’re pleased to inform you that your manuscript has been judged scientifically suitable for publication and will be formally accepted for publication once it meets all outstanding technical requirements.

Kind regards,

Andrew Sawers, PhD

Academic Editor

PLOS ONE

Additional Editor Comments (optional):

Reviewers' comments:

Reviewer's Responses to Questions

**Comments to the Author**

1. If the authors have adequately addressed your comments raised in a previous round of review and you feel that this manuscript is now acceptable for publication, you may indicate that here to bypass the “Comments to the Author” section, enter your conflict of interest statement in the “Confidential to Editor” section, and submit your "Accept" recommendation.

Reviewer #1: All comments have been addressed

Reviewer #2: All comments have been addressed

2. Is the manuscript technically sound, and do the data support the conclusions?

Reviewer #1: Yes

Reviewer #2: Yes

3. Has the statistical analysis been performed appropriately and rigorously? 

Reviewer #1: N/A

Reviewer #2: Yes

4. Have the authors made all data underlying the findings in their manuscript fully available?

Reviewer #1: Yes

Reviewer #2: Yes

5. Is the manuscript presented in an intelligible fashion and written in standard English?

Reviewer #1: Yes

Reviewer #2: Yes

6. Review Comments to the Author

Reviewer #1: The major and minor concerns communicated in the first review have been addressed clearly, professionally, and with support from the literature when needed.

Reviewer #2: The authors have addressed my comments and made consistent changes in the manuscript. The manuscript is now acceptable for publication.

7. PLOS authors have the option to publish the peer review history of their article (what does this mean?). If published, this will include your full peer review and any attached files.

Reviewer #1: No

Reviewer #2: **Yes: **Shamali Dusane

---

## [Editor Report · Acceptance letter]

30 May 2022

PONE-D-21-15719R1 

A qualitative photo-elicitation study exploring the impact of falls and fall risk on individuals with subacute spinal cord injury 

Dear Dr. Musselman:

I'm pleased to inform you that your manuscript has been deemed suitable for publication in PLOS ONE. Congratulations! Your manuscript is now with our production department. 

Kind regards, 

on behalf of

Dr. Andrew Sawers 

Academic Editor

PLOS ONE